# Synergistic Effects of La and Y on the Microstructure and Mechanical Properties of Cast Al-Si-Cu Alloys

**DOI:** 10.3390/ma15207283

**Published:** 2022-10-18

**Authors:** Luming Shuai, Xiuliang Zou, Yuqiang Rao, Xiaobin Lu, Hong Yan

**Affiliations:** 1Department of Material Processing Engineering, College of Advanced Manufacturing, Nanchang University, Nanchang 330031, China; 2Luxe Machinery (Gao’an) Co., Ltd., Gaoan 330800, China

**Keywords:** mixed rare earth, Al-Si-Cu alloy, microstructure, mechanical properties, thermodynamic calculation

## Abstract

The effects of La and Y on the microstructure and mechanical properties of cast Al-Si-Cu alloys were investigated by X-ray diffractometer (XRD), optical microscope (OM), and scanning electron microscope (SEM). The results indicated that the addition of La and Y had a great effect on the refinement of α-Al grains, the modification of eutectic Si phase, and the reduction of β-Al_5_FeSi length in Al-Si-Cu alloys. The A380 + 0.6 wt.% La/Y alloy exhibited the best microstructure and mechanical properties. The UTS and EI of the A380 + 0.6 wt.% La/Y alloy were 215.3 MPa and 5.1%, which were 22.9% and 37.8% higher than those of the matrix alloy, respectively. In addition, neither Al_11_La_3_ nor Al_3_Y generated by the addition of La and Y could not serve as the nucleation core of α-Al grains, so the grain refinement of α-Al originated from the growth limitation and constitutional supercooling. Since La and Y promote twinning generation and constitutional supercooling, the eutectic Si phase also changed from stripe-like to short fibrous or even granular and was significantly refined. Furthermore, thermodynamic calculations indicated that the Al_11_La_3_ phase was formed first and the Al_3_Y phase was generated on the Al_11_La_3_ phase.

## 1. Introduction

Al-Si alloys are widely used in automotive and aerospace industries due to excellent casting properties and high specific strength [1]. However, the traditional cast Al-Si alloys exhibited poor mechanical properties because of coarse α-Al dendrites, flake eutectic Si, and long needle-like β-Al_5_FeSi [2,3]. In order to improve the mechanical properties of cast Al-Si alloys, grain refinement and modification techniques have been widely studied and applied in recent years [4,5].

The chemical modification method has been shown to be very effective in extensive research. Rare earths can be used as both grain refiners and chemical modifiers, such as La [6], Y [7], Ce [8], Eu [9]. Especially, as the most economical rare earth metal, La has received much attention in the research. Related literature [10,11,12,13] investigated the modification effect of La on eutectic Si and β-Al_5_FeSi phase of Al-Si-Cu alloy. Mehdi et al. [10] reported that 0.2 wt.% La could reduce the amount of β-Al_5_FeSi phase and refine the size of α-Al grains in the alloy. Tsai et al. [11] indicated that La was not able to refine the Si phase well until the additions reached 1.0 wt.%. The highest fracture elongation of the modified alloy was achieved at 0.6 wt.% La addition. Moreover, according to the previous research [12,13], the degree of morphological alteration of microstructure by La was unsatisfactory because of the generation of needle-like La-rich compounds. Therefore, some researchers have focused on the modification effect of the combination addition of La with other elements on Al-Si alloys [14,15]. Qiu et al. [14] investigated the synergistic effect of La and Sr on the Al-Si alloy. It indicated that La not only limited the growth of eutectic Si, but also changed the morphology of the eutectic Si by entering the grooves of the eutectic Si. Cao et al. [15] reported the effects of La+Zr on Al-Si alloys. The results demonstrated that the heterogeneous core of α-Al was provided by Zr, which refined the α-Al grains and improved the tensile strength of the alloy. Meanwhile, La modified the eutectic Si phase and improved the fracture elongation of the alloy. In addition, Y also exhibited a powerful modification effect on eutectic Si. Liu et al. [16] reported that 0.2 wt.% Y could transform the morphology of Si phase from stripe to short fibrous or even granular. It also refined the α-Al grains and β-Al_5_FeSi phase at the same time. Li et al. [17] investigated the different effects of Fe, Mn, and Y on Al-Si alloys. It was shown that Y transformed the dendritic α-Al grains into equiaxed grains, and the grain size also became smaller by improving the nucleation supercooling. The enhancement of mechanical properties of the alloys with the addition of La or Y is mainly due to grain refining and the second phase strengthening. On the other hand, both La and Y could modify eutectic Si and refine α-Al grains, but there were some differences in the content used and the modification effect. Therefore, it would be interesting to investigate the effect of the combined addition of La and Y on the microstructure of Al-Si alloys. Previous literature [13,18] has reported the effect of La or Y alone in Al-Si alloys, respectively. However, there were few studies on the effect of combined La and Y additions on Al-Si alloys. Understanding the modification mechanism of La and Y on the microstructure of Al-Si alloys is contributing to the research of modifiers and broadening the applications of Al-Si alloys. However, the mechanism of the synergistic effect of La and Y in the Al-Si alloy is not clear.

This paper investigated synergistic effects of La and Y on the microstructure and mechanical properties of cast Al-Si-Cu alloys. In addition, the Gibbs free energy changes of Al_11_La_3_ and Al_3_Y produced by the reaction of La+Y and Al were also calculated. Meanwhile, the generation mechanisms of Al_11_La_3_ and Al_3_Y were analyzed based on the results of calculations and experiments. Furthermore, the mechanism of the microstructure evolution was discussed.

## 2. Experimental

### 2.1. Original Materials

The commercial A380 alloy was chosen as the matrix. The composition of the used A380 alloy is shown in Table 1.

### 2.2. Preparation of Master Alloys

To avoid the high burn rate of direct rare earth addition, the Al-Si-Cu alloy was modified with a homemade Al-10% La and Al-10% Y master alloys. Industrial pure Al (purity ≥ 99.87%), La, and Y were selected as raw materials to prepare rare earth master alloys by high-energy ultrasonic vibration method.

After the graphite crucible was preheated to 400 °C with the resistance furnace, the weighed and dried aluminum block was placed in the graphite crucible. At the casting temperature of 750 °C, La and Y were added with a tin foil wrapper. In order to obtain better dispersion [19], the preheated ultrasonic amplitude rod was probed into the molten liquid surface at 10–20 mm for intermittent ultrasonic vibration. The ultrasonic frequency was set to 20 kHz and the ultrasonic time was set to 15 min. The Al-10La and Al-10Y master alloys were obtained by pouring after ultrasonic completion.

### 2.3. Preparation of Modified Aluminum Alloys

The graphite crucible was placed in a resistance furnace and preheated to about 400 °C. The weighed lump of A380 alloy was put into the graphite crucible and heated up to 750 °C to melt it completely. Then, the previously prepared Al-10%La and Al-10%Y master alloys were added to the melt alternately to obtain rare earth aluminum alloy samples with different contents of La/Y (0 wt.%, 0.3 wt.%, 0.6 wt.%, 0.9 wt.%). Meanwhile, the melt was also treated with ultrasound for 15 min. After cooling to 720 °C, the melt was skimmed and poured into a preheated mold (300 °C), as shown in Figure 1a. During the entire experimental melting process, Argon gas was used to protect the melt from oxidation. The samples were taken from the bottom part of the casting after it cooled. The corresponding chemical compositions of the prepared alloys with different modifiers are shown in Table 2.

### 2.4. Microstructure Characterization

After all samples were etched by 0.5% HF solution, the microstructure and the fracture surface was observed by optical microscopy (OM, OLYMPUS BX51 microscope, Olympus Metrology, Inc., Tokyo, Japan) and scanning electron microscopy (SEM, Quanta 200, FEI Metrology, Inc., Hillsboro, OR, USA) equipped with energy dispersive spectroscopy (EDS, FEI Metrology, Inc., Hillsboro, OR, USA). The quantitative data analysis was also performed using the software IPP6.0. To determine the specific phases in the alloy specimens, X-ray diffractometer (XRD, D8 advance, Bruker, Inc., Karlsruhe, Germany) was used for the tests and the data obtained were analyzed using jade5 software.

According to the Chinese standard GB/T228-2002, these metal rods with different compositions were processed into tensile specimen bars with a diameter of 6 mm and a standard distance of 36 mm, as shown in Figure 1b. Tensile testing was performed on an electronic universal tensile test machine, model SUNS UTM5105. The starting pitch of the samples was measured at 50 mm and the tensile rate was set at 1 mm/min. To avoid errors in the experiments, five sample bars were selected from each tensile sample bar.

## 3. Results

### 3.1. Effect of Adding La+Y on the Microstructure of the Alloy

Figure 2 shows the OM images of matrix alloys with different contents of La+Y. As indicated by the arrow in Figure 2a, the matrix alloy is mainly composed of coarse α-Al grains, striped eutectic Si phase, and long needle-like β-Al_5_FeSi phase. In contrast, the addition of La and Y had a significant effect on modification of the cast microstructure. As exhibited in Figure 2b–d, the morphology of the eutectic Si phase in the alloy changes from strip-like to short fibrous or even granular. Meanwhile, both α-Al grains and β-Al_5_FeSi phase are well refined. The secondary dendrite arm spacing (SDAS) of α-Al grains, the area of Si phase, and the length of β-Al_5_FeSi phase were measured by IPP software, as shown in Figure 3. The results show that the A380 + 0.6 wt.% La/Y alloy exhibits the best refinement of microstructure. The SDAS and the length of β-Al_5_FeSi is decreased from 233 μm and 77.56 μm to 58 μm and 24.95 μm, respectively. The area of eutectic Si is also decreased from 289 μm^2^ to 6 μm^2^. Moreover, Figure 2c,d exhibit similar microstructure rather than improvement, which indicate that the increase of La+Y is not in favor of the microstructure optimization of the alloy anymore.

Figure 4 shows the SEM images of matrix alloys with different (La+Y) contents. The gray long needle-like phase and bright spot-like phase are observed in Figure 4a, which are inferred to be β-Al_5_FeSi phase and Al_2_Cu phase by combining with EDS and XRD analysis (Figure 5). As shown in Figure 4b, some bright short rod-like phases are also observed, which are inferred to be Al_11_La_3_ and Al_3_Y phases in combination with EDS analysis and corresponding binary phase diagram [20,21]. Comparing the EDS of Fe with La and Y, as shown in Figure 4a–d, it can be seen that the rare earth phase of the alloy does not contain Fe elements. So, the red circle in Figure 4b shows some rare earth phases attached to the needle-like β-Al_5_FeSi phase. According to the previous research [22], the rare earth elements can hinder its growth by attached to the β-Al_5_FeSi phase. Moreover, some letter-like α-Al_8_Fe_2_Si phases can be observed in Figure 4c. The mechanical properties of the alloy are much less impaired by the α-Al_8_Fe_2_Si phase compared to the needle-like β-Al_5_FeSi phase. In addition, as shown in Figure 4d, the bulk gray La-rich phases are observed in the A380 + 0.9 La/Y alloy, which can affect the mechanical properties of the alloy. On the other hand, the equilibrium partition coefficient of copper atoms is reduced because the enriched La and Y in the solid–liquid interface front hinders the diffusion of Cu atoms. Therefore, the solubility gradient of Cu in the liquid phase at the interface front increases, which increases the compositional supercooling of the alloy. So, the intensity of the peaks from the Al_2_Cu phase noticeably decreases with the addition of alloying elements, which is consistent with the XRD (Figure 5).

### 3.2. Mechanical Properties

The mechanical properties of matrix alloys with different contents of La+Y are tested, as shown in Figure 6. The ultimate tensile strength (UTS) and elongation index (EI) of the matrix alloy are 175.2 MPa and 3.7%, respectively. The A380 + 0.6 wt.% La/Y alloy has the best UTS and EI, which reach 215.3 MPa and 5.1%, respectively. These represent an increase of 22.9% and 37.8%, respectively, compared to the base alloy. However, the UTS and EI start to decrease when the addition of La+Y reached 0.9 wt.%.

The mechanical behavior of the alloy is controlled by the microstructure of the alloy. The addition of 0.6 wt.% La+Y results in a significant refinement of the α-Al grains and Si phase, which affect the UTS and EI of the alloy. When the contents of La+Y is excessive, the rare earth phase becomes coarse. The coarse phase causes concentration of stress, which adversely affects the tensile strength and elongation of the alloy.

### 3.3. Fracture Analysis

Figure 7 shows the SEM images of the fractures of the matrix alloys with different La+Y additions. As indicated by the arrow in Figure 7a, the fracture surface of the unmodified alloy is distributed with large cleavage platform and a few dimples, so brittle fracture is the main mode of fracture of the unmodified alloy. However, La+Y leads to a trend of fracture mode alteration from brittle fracture to ductile fracture. When the addition of La+Y reaches 0.6 wt.%, as shown in Figure 7c, more dimples and few cleavage platforms can be observed on the fracture surface, so ductile fracture becomes the main fracture mode of the alloy at this time, which is beneficial to the improvement of the mechanical properties of the alloy. However, when the rare earth addition reaches 0.9 wt.%, large cleavage platforms appear at many places, as shown in Figure 7d, which are typically brittle fractures. According to the SEM image (Figure 4d), rare earth phases form agglomerates, which may become a source of fracture in the alloy and damages the mechanical properties of the alloy, which is consistent with the results in Figure 6.

Figure 7e shows an enlarged image of the fracture in the A380 alloy with 0.6 wt.% La+Y additions. The elemental distributions of point 1 and point 2 are obtained by point scanning. The addition of 0.6 wt.% La+Y results in a small-sized massive Fe-rich phase (point 2), which limits the impairment of the relative mechanical properties by the long needle-like β-Al_5_FeSi in the alloy. Point 1 is a rare earth phase, which is small enough to benefit the mechanical properties of the alloy as well.

The Si phase is an important factor affecting the fracture mechanism of Al-Si alloys [23]. The coarse needle-like Si phase promotes stress concentration and distributes all over the alloy. The coarse Si phase is brittle, which can crack and break away from the matrix. When the alloy is subjected to external forces, the needle-like eutectic Si phase will rotate, resulting in internal stress concentration around the eutectic Si phase. When the concentrated internal stress exceeds the stress required for fracture, the Si phase causes the alloy to fracture [24]. After the addition of La+Y, the Si phase is obviously refined and the mechanical properties are improved.

## 4. Discussion

### 4.1. Effect of La and Y on α-Al Grains

During solidification, La and Y are enriched at the solid–liquid interface front due to the low solid solution of La and Y in the Al-Si-Cu alloy. Moreover, the La and Y are surface active elements, which make the surface tension and critical nucleation radius of the solution smaller [14]. Meanwhile, La and Y form a surface-active film between the grains and the solution, which inhibits the grain growth and refines the grains, as shown in Figure 8.

On the other hand, compositional supercooling is generated through the aggregation of La and Y elements at the solid–liquid interface fronts during the solidification of α-Al primordial crystals. The solid–liquid interface changes from flat interface to regular interface and irregular interface due to the compositional supercooling during solidification. Without considering convection, the presence of compositional supercooling in solution can be determined by the following equation [12]:(1)$$\frac{G_{L}}{v}\leq \frac{m_{L}C_{0}\left(1-k_{0}\right)}{D_{L}k_{0}}$$
(2)$$k_{0}=c_{S}^{*}/c_{L}^{*}$$
where G_L_ is the temperature gradient at the solid-liquid interface front (°C∙mm^−1^), m_L_ is the slope of the liquid phase line (°C∙(%∙mm)^−1^), C_0_ is the original composition of solute in the alloy (%), k_0_ is the solute partition coefficient, D_L_ is the solute diffusion coefficient, v is the solidification rate, $c_{S}^{*}\;$is the solute solubility in the solid phase, $c_{L}^{*}$ is the solute solubility in the liquid phase.

From Equations (1) and (2), it can be seen that the k_0_ is small because of the low solubility of La and Y in aluminum, so that the compositional supercooling is easily formed during the solidification of the alloy. The nucleation rate of the alloy is directly proportional to the supercooling degree, with the larger the supercooling degree, the higher the nucleation rate because of the lower nucleation work required for nucleation. In the case of supercooling, the increase in C_0_ causes an increase in $c_{L}^{*}$ and thus k_0_ decreases, which leads to the increase in compositional supercooling resulting in a change in the solidification interface morphology. Especially, the compositional supercooling zone at the interface front will gradually widen as C_0_ increases with the decrease of the ratio of temperature gradient to solidification rate G_L_/v and the enrichment of rare earths during the solidification process. If the compositional supercooling is higher than the supercooling degree required for nucleation of the effective substrate for heterogeneous nucleation, a large number of free equiaxed crystal nuclei may be generated directly in the compositional supercooling zone. Therefore, compositional supercooling can increase the nucleation rate and refine the grains. In addition, the compositional supercooling also changes the dendrite morphology to smaller-sized equiaxed crystals. This is because the deeper supercooling causes the melting of the dendrite roots during the growth of dendrites, resulting in the presence of a large number of free nuclei, which changes the growth of dendrites, and the transformation of α-Al grains from developed dendrites to smaller-sized equiaxed crystals.

Figure 9 show the DSC curves of the A380 alloys at a 10 °C/min heating rate. Peak 1 corresponds to the melting of eutectic Si, and peak 2 corresponds to the melting of primary α-Al. Table 3 shows the results of the analysis of the DSC curves. Where ΔT_1_ is the nucleation supercooling of eutectic Si and ΔT_2_ is the nucleation supercooling of primary α-Al. The results show that La and Y can increase ΔT_1_ and ΔT_2_, which improve the nucleation rate. Finally, the grains in the alloy are refined.

### 4.2. Modification of Eutectic Si Phase by La and Y

In the study of modifiers for Al-Si alloys, the modification of eutectic Si phase is the focus of modifier initiation [25]. The modifier atoms generally take effect by adsorption on the eutectic Si growth platform. Depending on the adsorption location, two types of modification effects are possible [26]: (i) by adsorption along the re-entrant edge, thereby preventing further attachment of Si atoms to the growing crystal and growth in that direction, and (ii) by adsorption on the Si{111} growth platform and promoting a shift in the Si phase stacking order. The first possibility is a generalization of the poisoning case of the twin plane re-entrant mechanism, while the latter possibility is a modification effect of the impurity-induced twinning theory (IIT) mechanism. The adsorption of modified atoms on the {111} step surface of Si and the changes in the stacking order promote the frequent formation of twins.

According to the IIT [27], if the ratio of the atomic radius of La/Y and Si is 1.646, then La/Y can induce twin crystal formation on the Si phase. The ratio of La, Y atoms to Si atoms is 1.59 and 1.55, respectively, which is very close to 1.646. Therefore, the elements La and Y can promote the twinning phenomenon. A large number of twin crystals are generated on the Si phase growth platform, resulting in a shift from anisotropic to isotropic growth of the Si phase. The morphology of eutectic Si is transformed from slat-like to fibrous. In addition, the modifier makes the Si crystal decelerate in the original growth direction, so the Si phase is no longer growing faster than the α-Al phase [28]. As shown in Figure 2c, the eutectic Si is transformed from long strips to short fibers and partially spheroidized. The size of the eutectic Si is significantly reduced and no longer seriously damages the mechanical properties of the alloy.

In addition, according to Hume–Rothery criterion [29], the solid solution will be very small when the difference of atomic radii is more than 15%. Therefore, rare earth elements will be gathered at the solid–liquid interface front during solidification. The radii and relative atomic masses of La and Y atoms are very different from those of Al and Si atoms. It is difficult for La and Y atoms to enter the lattice of α-Al and eutectic Si. Therefore, the atoms of La and Y distribute on the Si surface and dendrite surface at the front of the solid–liquid interface can act as a hindrance to the growth of Si phase and α-Al dendrites, as shown in Figure 8. As a result, solute diffusion and exchange will be inhibited, leading to a more moderate accumulation of solutes on the Si surface. Finally, the growth rate of eutectic Si is slowed down and the coordinated growth of α-Al and Si is achieved.

### 4.3. Effect of La and Y on Fe-Rich Phase

In Al-Si-Cu alloys, the modifiers that are effective for the Si phase usually work well for β-Al_5_FeSi as well. La and Y will also be adsorbed on the surface of β-Al_5_FeSi after entering the melt. This phenomenon will result in the formation of atomic films that inhibit the diffusion and precipitation of Fe atoms, which finally limited the growth of the β-Al_5_FeSi phase [30]. In addition, according to Chen et al. [22], the addition of La generates the Fe_9_LaSi_4_ phase in the Al-Si-Cu alloy, which consumes Fe elements. On the other hand, a part of Y also reacts with Si to form the Al-Si-Cu-Y phase [31], which attaches to β-Al_5_FeSi, consuming the atoms of nearby Si and hindering its growth, as in Figure 4b. Moreover, the length of the β-Al_5_FeSi phase decreases due to the restrained and consumed of Fe atoms in the melt, which could be observed from the Fe distribution of the EDS surface scan in Figure 4a–d. The reduction of the length of β-Al_5_FeSi phase and the appearance of bulk α-Al_8_Fe_2_Si phase in the alloy will reduce the negative impact of the original long needle-like β-Al_5_FeSi on the mechanical properties. 

### 4.4. Thermodynamic Analysis of Rare Earth Phase Generation

According to the binary phase diagrams of Al-Y and Al-La, Al_3_Y [20] and Al_11_La_3_ [21] are produced in the alloy, with melting points of 637 °C and 628 °C, respectively. Combined with the analysis of EDS and XRD, the fine rod-like bright phases in the modified alloy can be recognized as Al_11_La_3_ and Al_3_Y formed by the reaction of La and Y with Al. In order to analyze the generation capacity of Al_11_La_3_ and Al_3_Y, the Gibbs free energy of the two phases can be obtained using classical thermodynamic analysis [32].

According to the above discussion, Al_11_La_3_ and Al_3_Y are generated in the alloy with the following reaction equations:3Al + Y = Al_3_Y,(3)
11Al + 3La = Al_11_La_3_.(4)

According to Kirchhoff’s law, the relationship between the standard enthalpy of production and the isobaric heat capacity in a chemical reaction is as follows [33]:(5)$${\left[\frac{\partial \left({\Delta H}_{T}^{0}\right)}{\partial T}\right]}_{p}=\Delta c_{p}$$

Under isobaric conditions, the equation can be transformed into:(6)$${\Delta H}_{T}^{0}={\Delta H}_{T_{0}}^{0}+\int_{T_{0}}^{T}{\Delta c}_{p}\mathrm{dT}$$

The standard entropy of the reaction can be calculated using the following equation:(7)$$\Delta S_{T}^{0}=\Delta S_{T_{0}}^{0}+\int_{T_{0}}^{T}\frac{\Delta c_{p}}{T}\mathrm{dT}$$

Thus, the standard free energy of production for a pure matter reaction can be expressed as:(8)$$\Delta G_{T}^{0}=\Delta H_{T_{0}}^{0}-T\Delta S_{T_{0}}^{0}+\int_{T_{0}}^{T}{\Delta c}_{p}\mathrm{dT}-T\int_{T_{0}}^{T}\frac{\Delta c_{p}}{T}\mathrm{dT}$$
where T_0_ is the reaction starting temperature and T is the reaction ending temperature. The Gibbs free energy of Reaction (3) and Reaction (4) with respect to temperature T can be obtained from Equation (8), as shown in Table 4.

From the results, the Gibbs free energy change is negative for both Al_11_La_3_ and Al_3_Y in the experimental temperature range of 500 to 780 °C so that the reaction can proceed spontaneously due to the decrease in the total energy of the system during the reaction. The stability and formation ability of Al_11_La_3_ are much higher than that of Al_3_Y due to the large difference between the Gibbs free energy change of Al_11_La_3_ and Al_3_Y. Therefore, at the solid–liquid interface front during solidification, the Al_11_La_3_ phase should be formed by the reaction of La with Al first. Then, the Al_3_Y phase should be formed by the reaction of Y with Al. Al_11_La_3_ and Al_3_Y are aggregated at the grain boundaries due to the low solubility of La and Y in Al and Si. So, Al_3_Y is attached to the Al_11_La_3_ phase to generate because of the sequence of generation, which is consistent with that shown in Figure 4.

### 4.5. Effects of La and Y on the Mechanical Properties of A380 Alloys

La, Y can improve the mechanical properties of A380 alloy, as shown in Figure 6. The strengthening mechanism is dominated by grain refinement and the second phase strengthening. Grain refinement leads to more grain boundaries, which increase the resistance to dislocation movement and improve the resistance to deformation [34]. The contribution of grain refinement on mechanical properties can be measured by the Hall–Petch formula [35]:(9)$$\Delta \sigma_{\mathrm{Hall}-\mathrm{Petch}}={\mathrm{Kd}}^{-1/2}$$
where Δσ_Hall-Petch_ is the yield strength increment, K is a parameter on influence extent of grain boundaries to yield strength, and d is the average grain size. Therefore, grain refinement, which means a decrease in d-value, will lead to an improvement in mechanical properties. 

In addition, Al_3_Y and Al_11_La_3_ in the alloy will also provide second-phase strengthening by hindering dislocation movement in tensile experiments. The second phase strengthening in an alloy can be measured by the Orowan mechanism with the following equation [36]:(10)$$\Delta \sigma_{P}=\frac{2\mathrm{MGb}}{1.18\times 4\pi \times \left|\lambda -d\right|}\ln\frac{d}{2b}$$
where M is the Taylor factor, G is the shear modulus of α-Mg, b is the Burgers vector, λ is the particle interspacing, d is the particle size. The interaction of Al_3_Y and Al_11_La_3_ with dislocations generates dislocation loops through the Orowan bypassing mechanism, which hinders the dislocation movements in the alloy and reinforces the mechanical properties.

## 5. Conclusions

The effects of La and Y on the microstructural evolution and mechanical properties of cast Al-Si-Cu alloys were investigated in this work. The main conclusions drawn from this work are as follows:

La+Y can be used as an effective modifier for Al-Si-Cu alloys. With the optimum content of 0.6 wt.%, the alloy has the best microstructure. The coarse dendritic primary α-Al grains are fully refined and transformed to equiaxed grains. Meanwhile, the eutectic Si is modified from strips to short fibrous or even granular. In addition, the length of long needle-like β-Al_5_FeSi is reduced and bulk α-Al_8_Fe_2_Si is observed.

The refinement effect of La and Y on the microstructure is mainly derived from the phenomenon of compositional supercooling and the limitation of solute diffusion. Meanwhile, La and Y can promote the generation of twins on the Si phase and modify the morphology through the IIT mechanism.

The Gibbs free energy changes of the generated Al_11_La_3_ phase and Al_3_Y phase are both negative in the alloys. Al_11_La_3_ has more negative Gibbs free energy change than the Al_3_Y phase, thus Al_11_La_3_ is more stable. At low La+Y content, Al_3_Y is generated on Al_11_La_3_ and covers it. However, the uncovered Al_11_La_3_ is observed with an La+Y content of 0.9 wt.% in the alloy.

The addition of 0.6 wt.% La+Y result in UTS and EI of 215.3 MPa and 5.1%, respectively. These values increase by 22.9% and 37.8%, respectively, compared to the unmodified alloy. The fracture test reveals that the main fracture mode is changed from brittle fracture to ductile fracture.

## Figures and Tables

**Figure 1 materials-15-07283-f001:**
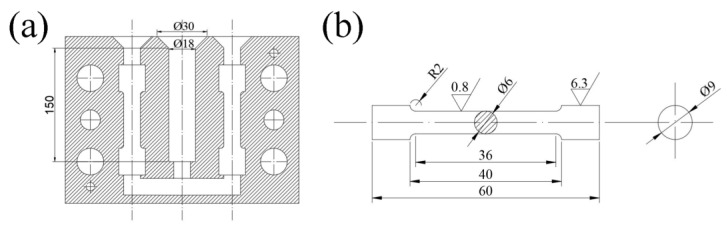
(**a**) mold drawing (unit: mm) and (**b**) tensile specimen bar (unit: mm).

**Figure 2 materials-15-07283-f002:**
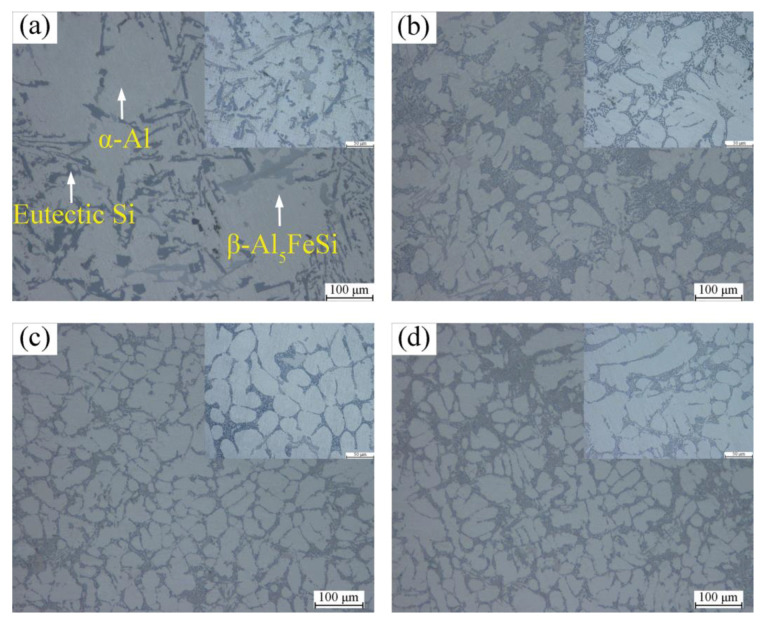
High magnification OM images of cast A380 alloys with different (La+Y) contents: (**a**) unmodified alloy; (**b**) 0.3 wt.%; (**c**) 0.6 wt.%; (**d**) 0.9 wt.%.

**Figure 3 materials-15-07283-f003:**
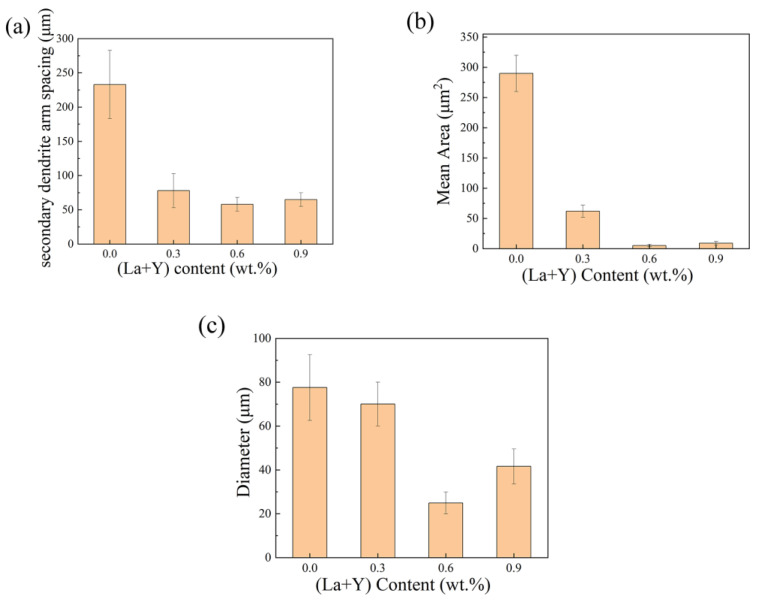
(**a**) The SDAS of α-Al grain, (**b**) the area of eutectic Si, and (**c**) the length of β-Al_5_FeSi phase of A380 alloys with different (La+Y) contents.

**Figure 4 materials-15-07283-f004:**
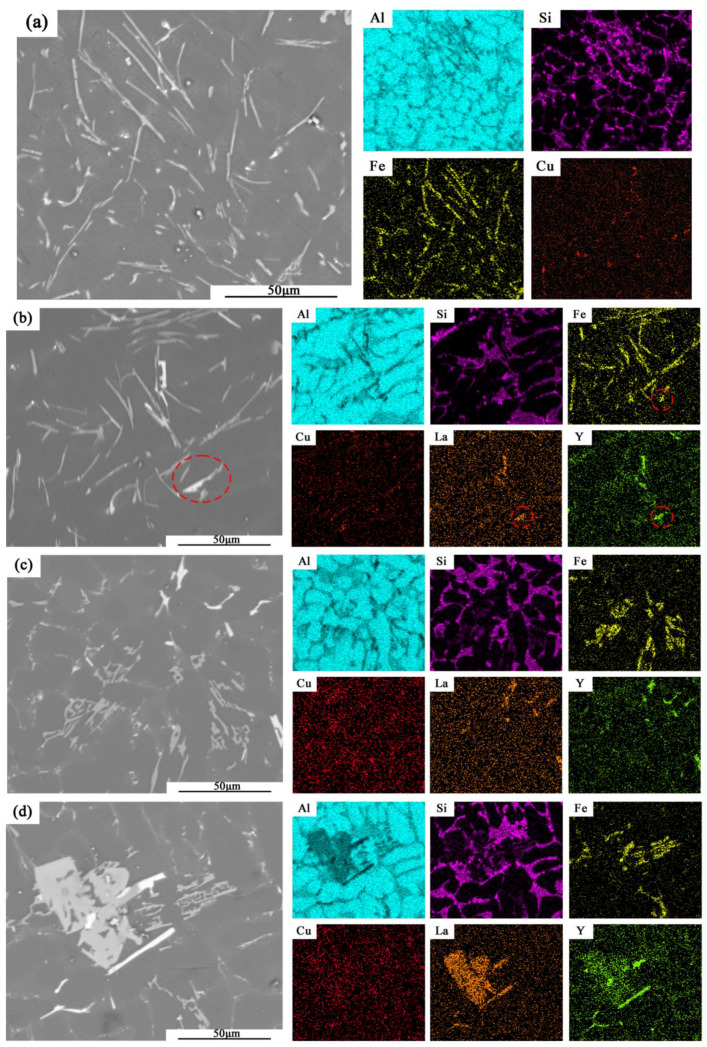
Scanning electron micrographs of cast A380 alloys with different (La+Y) additions: (**a**) matrix alloy; (**b**) 0.3 wt.%; (**c**) 0.6 wt.%; (**d**) 0.9 wt.%.

**Figure 5 materials-15-07283-f005:**
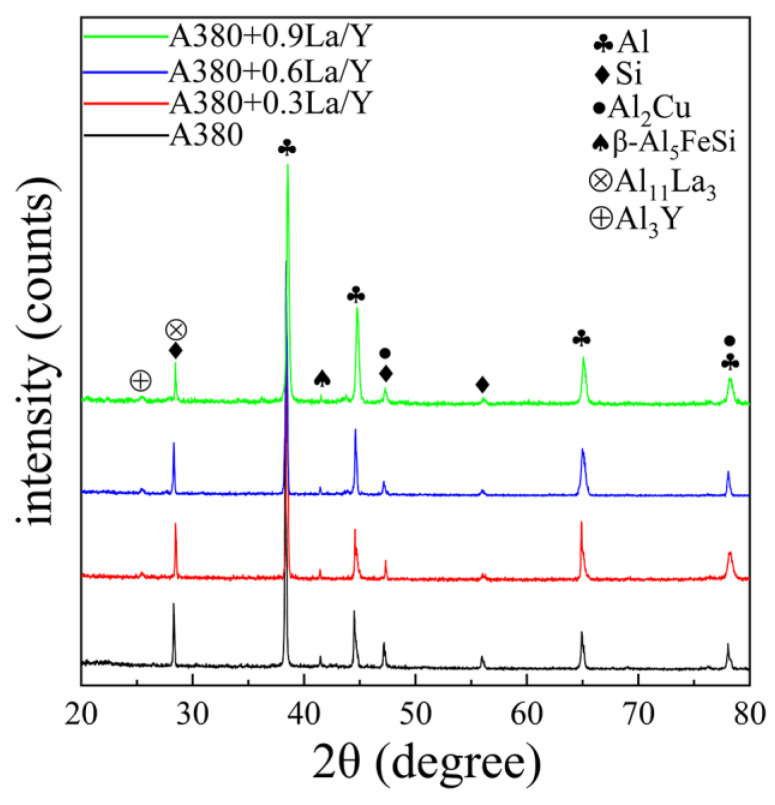
XRD patterns of A380 alloys with different (La+Y) contents.

**Figure 6 materials-15-07283-f006:**
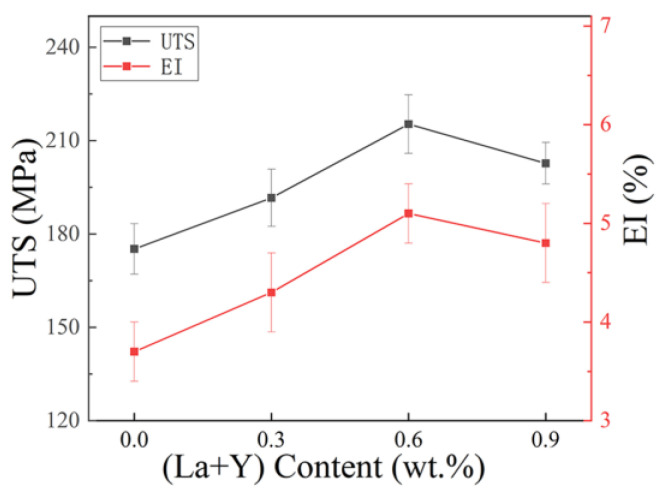
Mechanical properties of A380 alloys with different (La+Y) contents.

**Figure 7 materials-15-07283-f007:**
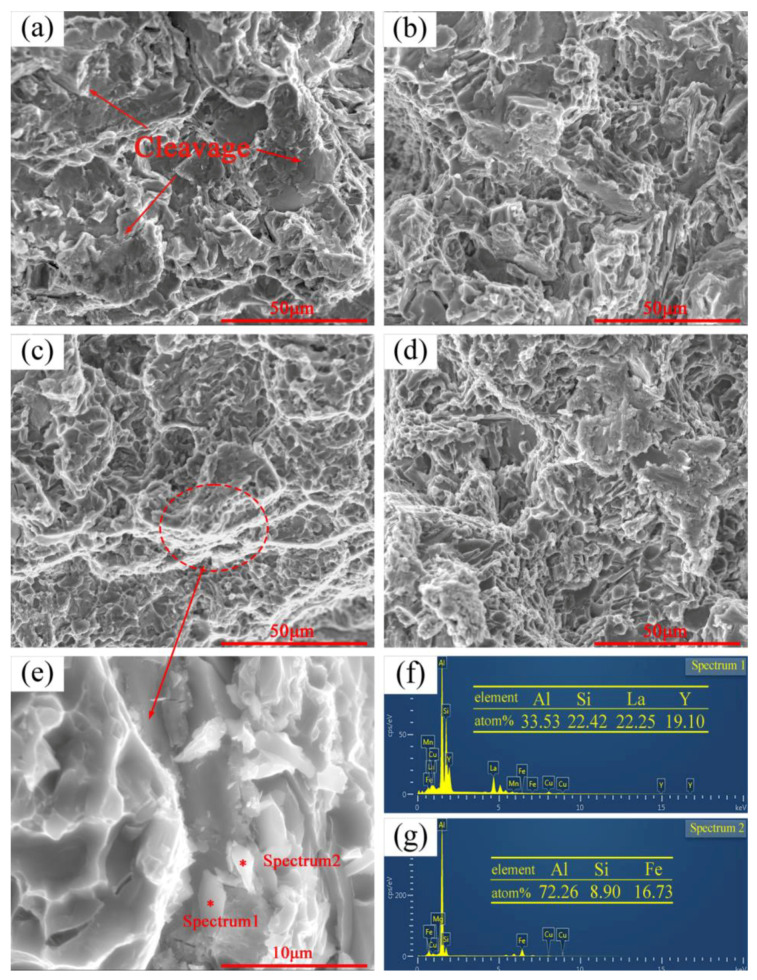
SEM of fractures of cast A380 alloys with different (La+Y) contents: (**a**) matrix alloy; (**b**) 0.3 wt.%; (**c**) 0.6 wt.%; (**d**) 0.9 wt.%; (**e**) enlarged view of the selected area in (**c**); (**f**) elemental point scan at point 1; (**g**) elemental point scan at point 2.

**Figure 8 materials-15-07283-f008:**
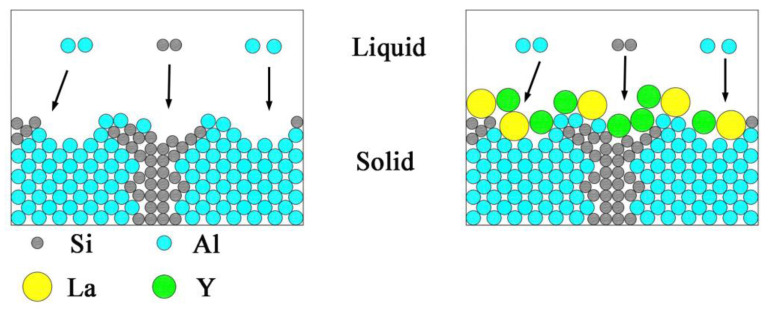
Limitation of the growth of α-Al and eutectic Si phases by La and Y.

**Figure 9 materials-15-07283-f009:**
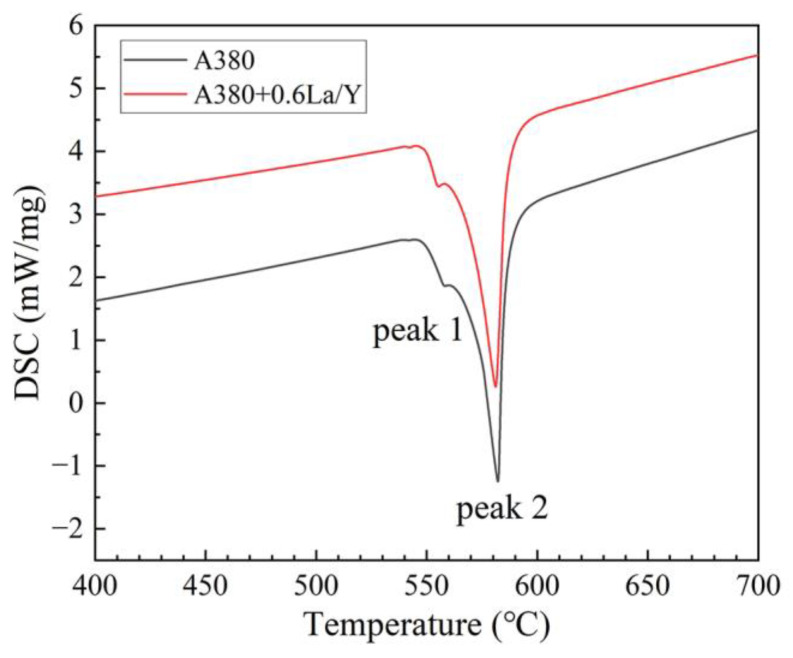
DSC curves for the La/Y-free and La/Y-contained A380 alloys.

**Table 1 materials-15-07283-t001:** Composition of A380 alloy.

Elements	Si	Cu	Mg	Zn	Fe	Mn	Al
(wt.%)	10.22	2.40	0.86	0.23	1.21	0.27	Bal.

**Table 2 materials-15-07283-t002:** Chemical compositions of experimental alloys (wt.%).

Alloys	Si	Cu	Mg	Fe	La	Y	Al
A380 + 0 La/Y	10.22	2.40	0.86	1.21	0	0	Bal.
A380 + 0.3 La/Y	10.14	2.39	0.82	1.22	0.16	0.14	Bal.
A380 + 0.6 La/Y	10.09	2.39	0.83	1.18	0.33	0.31	Bal.
A380 + 0.9 La/Y	10.01	2.38	0.84	1.17	0.42	0.46	Bal.

**Table 3 materials-15-07283-t003:** The Results of DSC Analysis.

No.	Eutectic Si		Primary α-Al	
	Peak 1 (°C)	ΔT_1_ (°C)	Peak 2 (°C)	ΔT_2_ (°C)
A380	556.50	47.30	582.25	21.55
A380 + 0.6 La/Y	555.43	50.97	581.14	25.26

**Table 4 materials-15-07283-t004:** Gibbs free energy of Al_3_Y and Al_11_La_3_.

Reaction	3Al + Y = Al_3_Y	11Al + 3La = Al_11_La_3_
${\Delta G}_{T}^{0}$ (J∙mol^−1^)	−185,600 + 83.89 T	−574,001 + 64.313 T

## Data Availability

The data presented in this study are available on request from the corresponding author.

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
