# Peer review of "Synergistic Effects of La and Y on the Microstructure and Mechanical Properties of Cast Al-Si-Cu Alloys"

_materials, 2022, doi:10.3390/ma15207283_

Round 1

Reviewer 1 Report

 Authors attempted to study the synergistic effects of La and Y on the microstructure and mechanical properties of cast Al-Si-Cu alloys. It is a well written manuscript but requires moderate revision. The review suggestions are appended below.
1) The introduction section needs to be more elaborate. What are the strengthening mechanisms of La and Y should be clearly described. It is necessary to highlight the solubility limit of La and Y in Al-Si-Cu alloy.
2) It would be nice if any phase diagram can be included to show the phase transition.
3) In the experimental section it was mentioned as ultrasonic assisted casting. However, from the EDS maps it is clear that reinforcements were not thoroughly mixed which indicates lack of stirring time or low frequency settings to cause cavitation. Please provide a proper reason in the results and discussion to promote better understanding to the readers. Please go through the following journals for better insights into the ultrasonic assisted casting process.
https://doi.org/10.1016/j.matlet.2020.127879
https://doi.org/10.1016/j.matlet.2020.129113
4) In manuscript the thermal conductivity, heat transfer coefficient, specific heat capacity and melting points of La and Y should be included for better clarity.
5) Cooling rate can be well explained using growth rate exponent. It would be nice if authors could calculate cooling rate based on SDAS spacing.
6) In XRD, peaks with very minor intensities of La and Y were present which could not be properly indexed Al3Y and All1 La3. It is because the melting point of La and Y are very high as compared to that of Al. Possibly there won't be any chemical reaction between them to form a new intermetallic compound. Please double check the results.
7) Addition of reinforcement materials having different thermal expansion coefficients and melting points can lead to inducement of lattice strains which can be seen in terms of peak shifting. Did authors observe such changes in XRD? Moreover, it is good to correlate the grain refinement with peak broadening. Are there any such observations?
8) It is necessary to include some fundamental insights such as how Zener Pinning takes place. How the grain nucleation is ceased. Please look at the following journal for deep insights. How much would be the change in undercooling temperature due to heterogeneous nucleant addition.
9) The tensile properties of reinforcement added samples could be well explained with the Orowan strengthening mechanism and classical Hall Petch empirical relationship. For better clarity refer to these following journals.
https://doi.org/10.1007/s12666-022-02756-6
https://doi.org/10.1016/j.matpr.2022.05.469
https://doi.org/10.1016/j.matlet.2021.130936
https://doi.org/10.1007/s41403-021-00269-0
https://doi.org/10.1016/j.matlet.2020.128578
10) Are there any porosities were observed with respect to the fraction of reinforcement addition?
11) In Al composite brittle fractures are very arguable. Minor addition of reinforcements won't be that effective in changing the dimple feature to a brittle fracture morphology. Please double check.       

Reviewer 2 Report

1) "To avoid the high burn rate of direct rare earth addition, the Al-Si-Cu alloy was mod- 78 ified with a homemade Al-10% La and Al-10% Y intermediate alloy. Industrial pure alu-79". Such alloys are called "master alloys"

2) In Figure 4, I really want to see the spectra of Cu. It is possible that there are phases with copper in the structure, as shown in "The Effect of Yittrium and Zirconium on the Structure and Properties of the Al–5Si–1.3Cu–0.5Mg Alloy"

3) It is also not said about the redistribution of copper in a solid solution. The intensity of the peaks from the Al2Cu phase noticeably decreases with the addition of alloying elements.

4) Please describe what was the sample for XRD.

5) It is necessary to provide data on the study of the properties of alloys after heat treatment, because alloys of this group are not used in industry in the cast state.

Reviewer 3 Report

1.      Authors must revise the manuscript for the grammatical and typo errors.

2.      Some of the figures seems to be adopted from the published sources, If so, it is recommended to submit the copyright forms.

3.      Line 130 and 131 is confusing, please revise it for better understanding.

4.      Figure 4 (d) SEM, what is the gray colour disturbance inside the SEM microstructure?

5.      Authors discussed that, there is a formation of β-Al5FeSi in the microstructure, but not reflected in XRD. Why?

6.      As per XRD, the 0.3% addition of La and Y showed refined microstructure compared to others. What is reason form grain refinement compared to parental alloy?

Round 2

Reviewer 1 Report

Accept

Reviewer 2 Report

Accept in present form